# Theory-Based Determinants of Stopping Drowsy Driving Behavior in College Students: A Cross-Sectional Study

**DOI:** 10.3390/ijerph21091157

**Published:** 2024-08-30

**Authors:** Md Sohail Akhter, Sidath Kapukotuwa, Chia-Liang Dai, Asma Awan, Omolola A. Odejimi, Manoj Sharma

**Affiliations:** 1Department of Social and Behavioral Health, School of Public Health, University of Nevada, Las Vegas, NV 89119, USA; akhtem1@unlv.nevada.edu (M.S.A.); asma.awan@unlv.edu (A.A.); manoj.sharma@unlv.edu (M.S.); 2Department of Teaching and Learning, College of Education, University of Nevada, Las Vegas, NV 89154, USA; chia-liang.dai@unlv.edu; 3Department of Educational Psychology, Leadership, and Higher Education, College of Education, University of Nevada, Las Vegas, NV 89154, USA; omolola.odejimi@unlv.edu; 4Department of Internal Medicine, Kirk Kerkorian School of Medicine at UNLV, University of Nevada, Las Vegas, NV 89102, USA

**Keywords:** college students, drowsy driving, fatigue-related accidents, multi-theory model, road safety

## Abstract

Drowsy driving among college students is a critical public health issue due to its significant impact on road safety. This cross-sectional study aimed to investigate the determinants of stopping drowsy driving behavior among college students using the multi-theory model (MTM) of health behavior change. Data for this study were collected from September to October 2023 via a 42-item psychometric valid, web-based survey disseminated via Qualtrics, involving 725 students from a large southwestern university. Nearly half of the participants (49.38%) reported drowsy driving in the past month. Hierarchical multiple regression analysis revealed that participatory dialogue (*p* = 0.0008) and behavioral confidence (*p* < 0.0001) significantly predicted the initiation of refraining from drowsy driving, with the final model explaining 36.4% of the variance. Similarly, emotional transformation (*p* < 0.0001) and practice for change (*p* = 0.0202) significantly predicted the sustenance of behavior change, with the final model accounting for 40.6% of the variance. These findings underscore the importance of targeted MTM-based interventions focusing on enhancing students’ awareness and confidence in managing drowsiness to mitigate drowsy driving, ultimately improving road safety and student well-being.

## 1. Introduction

Drowsy driving refers to the action of driving a motor vehicle when in a state of sleep or impeded by drowsiness, sleepiness, or exhaustion [1,2]. Symptoms of drowsy driving manifest as tiredness with repetitive yawning or blinking, impaired recollection of recent miles driven, failure to take appropriate exits, veering out of a lane, or colliding with a rumble strip [3]. Drowsy driving typically occurs when a driver gets little sleep; however, it can also be attributed to untreated sleep problems or working in shifts [4]. There are two main causes of drowsy driving, viz, (1) sleep restriction, which entails reducing the amount of sleep to allocate more time for work, study, socializing, or engaging in other activities; and (2) sleep fragmentation which causes insufficient sleep, having detrimental effects on an individual’s daytime performance [5]. Both prescription and over-the-counter drugs can induce drowsiness, and alcohol can exacerbate the effects of sleepiness, leading to increased impairment and drowsiness [6]. It presents a substantial public health and safety issue on roadways in both the United States and globally [7].

Research indicates that drowsy driving is alarmingly common in this population. For instance, a study reported that 16% of surveyed undergraduates had fallen asleep while driving, with 2% experiencing a motor vehicle accident due to sleepiness [8]. Robbins et al. (2021) highlighted that 20% of students reported having fallen asleep while driving [9]. The article by Lee et al. (2016) reports that 31.4% of the students reported driving while drowsy at least once in the past month and 71.9% at least once in their lifetime [10]. A study by Lindsay et al. (1999) found that 32% of college students reported experiencing a dozing and driving incident since starting college, underscoring the widespread nature of this issue [11]. A study conducted by Forquer et al. (2008) noted that 33% of students experience tiredness during the day, which contributes to impaired driving performance [12]. 

Drowsy driving among college and university students has significant consequences, including an increased risk of motor vehicle accidents due to impaired cognitive function and reaction times [9]. It negatively affects academic performance, as inadequate sleep is linked to reduced attention, memory, and cognitive processing speeds [13]. Additionally, chronic sleep deprivation from drowsy driving can lead to health issues like obesity, diabetes, cardiovascular disease, and weakened immune function [14]. It also compromises mental health, being associated with mood disorders such as depression and anxiety [8]. Furthermore, drowsy driving results in social and economic costs, including financial burdens from accidents and emotional tolls on families and communities [8].

The high prevalence of drowsy driving among college students can be attributed to several factors. Irregular sleep schedules and sleep deprivation are pervasive issues in this demographic, largely due to the demanding nature of college life. Hershner and Chervin (2014) highlighted that 70% of college students have insufficient sleep, significantly impacting their driving performance [14]. Beck et al. (2018) found that insufficient sleep, coupled with various professional, academic, and social demands, leads students to drive even when they are aware of their drowsiness [13]. 

Several studies have explored the specific determinants of drowsy driving among college students. Irregular sleep schedules, a common issue in this demographic, significantly contribute to the problem. Taylor and Bramoweth (2010) noted that the transition to college life often leads to reduced parental supervision, new social opportunities, and challenging academic demands, all of which disrupt regular sleep patterns and increase the likelihood of sleep deprivation. They also found that many students use substances such as medication, alcohol, and stimulants to manage their sleep and alertness, further exacerbating sleepiness and impairing driving abilities [8]. Similarly, Lindsay et al. (1999) highlighted the impact of sleep deficiency and irregular schedules, with nearly 58% of students involved in dozing incidents reporting that they had not completed a normal period of sleep before the incident and 56% acknowledging an irregular schedule in the preceding week [11].

The willingness to engage in drowsy driving is also influenced by students’ attitudes and perceptions. Lee et al. (2016) identified that perceived behavioral control and behavioral willingness were the strongest predictors of intentions to drive while drowsy among university students [10]. Students often underestimated the risks associated with drowsy driving, viewing it as a normal and unavoidable part of their lives. Williams et al. (2012) also noted that stress and academic pressures contribute to irregular sleep patterns, increasing the likelihood of drowsy driving [15]. Addressing these determinants through targeted interventions and educational campaigns is crucial for enhancing student safety and reducing the incidence of drowsy driving-related accidents. The findings from these studies underscore the need for improved sleep hygiene and awareness of the risks associated with drowsy driving. 

A range of interventions have been explored to address drowsy driving among college students. A distracted driving presentation significantly improved knowledge, attitudes, and behaviors in the short term [16]. Sleep education classes, online programs, and encouragement of naps have been suggested as viable interventions to decrease sleepiness and sleep deprivation [14]. However, college students often view drowsy driving as normal and uncontrollable and prefer messaging strategies that emphasize the consequences of drowsy driving [13]. The Theory of Planned Behavior (TPB) and the Prototype Willingness Model (PWM) have been suggested as useful frameworks for understanding and addressing drowsy driving because they offer insights into the factors that influence individuals’ intentions and willingness to engage in this risky behavior [10]. The TPB helps to explain how attitudes, perceived social pressures (subjective norms), and perceived control over the behavior contribute to one’s intention to drive while drowsy. The PWM adds an additional layer by considering the role of social prototypes and situational factors that can influence individuals to engage in drowsy driving even when they have no prior intention to do so. Together, these models provide a comprehensive approach to predicting and modifying drowsy driving behaviors [10]. Incorporating theoretical frameworks into initiatives addressing drowsy driving among college students is a beneficial approach. It aids researchers and practitioners in identifying relevant situational factors and procedures crucial for developing effective strategies to combat drowsy driving.

Therefore, the present study employs a modern, fourth-generation theoretical framework known as the multi-theory model (MTM) of health behavior change to investigate the antecedents of drowsy driving among college students [17,18]. The MTM separates behavioral change into two parts, including (1) initiation, which consists of three constructs: (a) participatory dialogue defined as the difference between the benefits and drawbacks of changing a behavior; (b) behavioral confidence defined as the individual’s certainty in fostering a behavioral change; and (c) changes in the physical environment regarding the support from the physical resources and (2) sustenance of behavior change, which also includes three constructs: (a) emotional transformation, which involves changes in an individual’s feelings regarding the behavioral change, self-affirmations, and goal setting; (b) practice for change is a construct that explains an individual’s ability to overcome obstacles while remaining focused on the change; and (c) changes in the social environment indicate the ability to maintain a behavior by receiving social support [17,18]. The MTM constructs are versatile in their application to health behaviors and have been utilized to elucidate a wide range of health behaviors among college students [19,20,21]. 

The innovative aspect of this study lies in the application of the MTM to drowsy driving behavior among college students. To the best of our knowledge, this is the first study to utilize the MTM in exploring this behavior within this population. By employing the MTM, this study not only addresses a critical public health concern but also demonstrates the versatility and applicability of this novel theoretical model in understanding and promoting behavior change in real-world settings. The MTM’s comprehensive approach offers new insights into both the initiation and sustenance of behavior change, making it a powerful tool for developing effective interventions. Thus, the purpose of this study is to elucidate the extent to which the MTM constructs of initiation and sustenance explain the cessation of drowsy driving behavior among college students at a large southwestern university. This study compared MTM scores between students who drove while drowsy and those who did not, without any special program or intervention to stop drowsy driving. The objective of this was to obtain a baseline understanding of how well the MTM differentiated these behaviors—even before any behavior change interventions are introduced. It was presumed that if the MTM is successful in differentiating them in this baseline circumstance, it would suggest that it can offer real promise for guiding intervention development in the future.

## 2. Materials and Methods

### 2.1. Study Design, Sample, and Study Participants

In the current study, a cross-sectional design was utilized. The sample size was calculated using G* Power [22]. An alpha of 0.05, power of 0.80, number of predictors of 14 (3 for constructs in each model and 11 for control variables), and an effect size of 0.03 yielded a sample size of 624. This was inflated by 15% for any missing responses, yielding a sufficient sample size of 718. The study utilized a cross-sectional methodology, gathering data from 23 September 2023 to 24 October 2023. It focused on students registered at a sizable university located in the southwestern region of the United States. Participants were required to be 18 years or older, fluent in English, and to have given their consent to take part in the study.

### 2.2. Ethical Considerations

The study was granted an exemption by the university’s institutional review board on 31 August 2023, under protocol number UNLV-2023-417. Consent from participants was obtained through their voluntary agreement to join the study. The informed consent form offered detailed insights into the study’s purpose, importance, potential risks, and the freedom to opt out at any point. Only those who chose the “Agree” option were allowed to move forward to the survey. Identifiable personal information, like names or email addresses, was not gathered.

### 2.3. Recruitment and Data Collection

Recruitment of participants began with announcements in the student newsletters of the university, followed by accessing student directory information from the registrar’s office to directly contact students via their university email addresses. These emails included thorough information about the research and an anonymous link to the survey. Upon clicking this link, participants were taken to an online survey hosted on Qualtrics (Provo, UT, USA). To ensure the integrity of the survey responses, the “Prevent multiple submissions” feature was enabled in Qualtrics to bar participants from submitting more than one response. Additionally, the bot detection feature was activated to screen out non-human respondents, and “RelevantID” was employed to identify and eliminate fraudulent responses.

### 2.4. Survey Instrument

The survey tool includes a 42-item questionnaire developed by the creator of the theory and refined through a two-stage expert panel review. A group of seven experts were invited to assess the questionnaire’s face and content validity through a two-round process. Two of the panel members were proficient in the field of public administration and motor vehicle services, two panel members had expertise working with college students, one expert was external to public health, and two members were content experts in health behavior research, health education, and instrument development and had knowledge of one or more theories or models. The independent experts were requested to assess the clarity, readability, and relevance of the questionnaire items. The wording of the items was modified slightly based on the comments from the experts. There were no items that were excluded from the questionnaire. All the experts agreed that the content and face validity of each of the MTM subscales were sufficient. The instrument achieved a Flesch reading ease score of 62.4 and a Flesch–Kincaid Grade Level of 6.0, indicating its readability. The questionnaire starts with a question to identify if the respondents have engaged in drowsy driving over the past 30 days. This is followed by six questions aimed at gathering information on alcohol consumption, incidents of driving under the influence (DUI), usage of prescription medication affecting driving ability, involvement in night shift work, experiences with sleep disorders, and attendance at late-night gatherings. The subsequent 30 items evaluate the components of the multi-theory model (MTM) related to both the initiation and maintenance aspects. The final five items are dedicated to collecting demographic information from the participants, including age, gender, race or ethnicity, academic year, and employment status.

Participatory dialogue in the study is evaluated through the perceived advantages and disadvantages, each assessed using five questions. These questions are rated on a five-point scale ranging from 0 (never) to 4 (very often), making the total possible scores for perceived advantages and disadvantages range from 0 to 20. The overall score for participatory dialogue is calculated by subtracting perceived disadvantages from perceived advantages, yielding a potential score range from −20 to 20. Behavior confidence is assessed using six items, with responses also recorded on a five-point scale (from 0 = not at all sure, to 4 = completely sure), resulting in a scoring range from 0 to 24 for behavior confidence. The other four constructs are each evaluated by three items, also using a five-point scale for responses, leading to a score range of 0 to 12 for each construct. The overall initial score is determined by a single item, utilizing a five-point scale (from 0 = not at all likely, to 4 = completely likely) that allows for a score range of 0 to 4. A similar approach is used to calculate the overall sustenance score.

### 2.5. Statistical Analyses

All data were analyzed using SAS version 9.4 (SAS Institute Inc.), Cary, NC. USA and MPlus version 8.5 (Muthén & Muthén, 2020), Los Angles, CA. USA.

For continuous variables, descriptive statistics were provided in the form of means and standard deviations, whereas frequencies and percentages were used to summarize categorical variables.

To evaluate the internal consistency of our scales, Cronbach’s alpha coefficients were computed for both the entire scale and its constituent subscales, which are delineated by specific constructs, to evaluate internal consistency. An alpha value of 0.70 was established as the minimum acceptable limit, signifying that the scale and its subscales possess an adequate level of reliability in measuring the constructs they are intended to assess.

To further explore reliability, unidimensionality, and validity, each scale was modeled as a latent variable using common factor analysis in Mplus. To test the construct validity, the scales turned factors were correlated, representing a measurement model. This was implemented using Mplus. A combination of the following fit indices was utilized to measure the degree of overall fit of the measurement model to the data: comparative fit index (CFI) and Tucker–Lewis Index (TLI) with values above 0.90 [23] and a root mean square error of approximation (RMSEA) and Standardized Root Mean Square Residual (SMRM) less than 0.08 [24]. We used the 0.10, 0.30, and 0.50 guidelines to explain the effect size for identifying small, medium, and large effects, respectively [25].

The study focused on two primary outcomes: the likelihood of intending to initiate refraining from drowsy driving and the likelihood of sustaining this behavior change among university students. The independent variables were based on the MTM constructs. Covariates included alcohol consumption, instances of DUI, use of prescription drugs affecting driving capabilities, engagement in night shift work, suffering from sleep disorders, and participation in late-night activities, along with demographic factors such as age, gender, race or ethnicity, academic standing, and employment status.

To identify mean differences in MTM constructs between students who had and had not engaged in drowsy driving in the preceding month, an independent samples t-test was applied. Hierarchical multiple regression and structural equation modeling were employed to predict the initiation and sustenance of quitting drowsy driving behavior. In the initiation model, covariates were entered first (Model 1), followed by the first construct of initiation (Model 2), the first two constructs (Model 3), and all constructs (Model 4). A similar stepwise approach was used for the sustenance model. This method assesses the incremental explanatory power of each predictor set while controlling for other variables [26]. We assumed that the final models of initiation and sustenance met assumptions of observation independence, linearity, normality, and homoscedasticity. These assumptions were verified through the Durbin Watson statistic for independence, partial regression plots for linearity, the Shapiro–Wilk test for normality, and the White test for homoscedasticity. The variance inflation factor (VIF) was used to assess multicollinearity in the final model. A significance level of 0.05 was maintained for all statistical tests. With the exception of the independent samples t-test, these models were also evaluated using the aforementioned fit indices.

## 3. Results

### 3.1. Sample Characteristics

A total of 725 valid responses were collected. Most of the participants were female students (66.21%, *n* = 480); there were 216 male students (29.79%), and 29 students (4.00%) who chose the “Other” option for gender (Table 1). The average age was 26.01 years, with a standard deviation of 9.20 years. There were 270 (37.24%) White; 178 (24.55%) Hispanic, Latino/a, or Latinx; and 132 (18.21) Asian or Asian American students. The remaining 145 (20.00%) students belonged to Multi-racial, Black, Native American, or other races/ethnicities (Table 1). Most participants were undergraduate students (66.20%, *n* = 480; Table 1).

### 3.2. Internal Consistency and Construct Validity

Table 2 displays the Cronbach’s alpha values along with their 95% confidence intervals (CIs) for all the scales utilized in the instrument. Each Cronbach’s alpha estimate exceeds the 0.70 threshold, affirming that the scales and subscales designed for measuring initiation and sustenance processes maintain acceptable levels of internal consistency.

Most of the fit indices utilized to measure the degree of overall fit of the initiation model met the conventional threshold of an acceptable level. The estimated indices were an RMSEA of 0.08, SRMR of 0.06, and CFI of 0.90, while the TLI of 0.88 was close to the cutoff value. We observed that the standardized factor loadings ranged from 0.40 to 0.93 (Figure 1).

These effects indicated that the initiation to the safe driving scale provided valid measurement of its constructs (i.e., perceived advantages, perceived disadvantages, behavioral confidence, and changes in the physical environment). We found advantages, behavioral confidence, and changes in the physical environment had small to moderate positive direct and significant effects on the initiation of safe driving behavior (i.e., ranging from 0.08 to 0.58).

This meant that as the emotional transformation and changes in the social environment increased, the sustenance of safe driving behavior increased as well. No significant effect was found of practice for change on the sustenance of safe driving behavior. We also found significant relationships among factors of emotional transformation, practice for change, and changes in social environment.

For the sustenance model, the fit of the model was excellent (i.e., CFI 0.99, TLI 0.94, RMSEA 0.05, and SRMR 0.02). The standardized factor loadings in the sustenance scale ranged from 0.61 to 0.92 (Figure 2). These effects indicated that the sustenance scale provided valid measurement of its constructs (i.e., emotional transformation, practice for change, and changes in social environment). We observed that the emotional transformation and changes in social environment had small to moderate and significant effects on the sustenance of safe driving behavior (i.e., 0.50 and 0.17 respectively).

This meant that as the emotional transformation and changes in the social environment increased, the sustenance of safe driving behavior increased as well. No significant effect was found of practice for change on the sustenance of safe driving behavior. We also found significant relationships among factors of emotional transformation, practice for change, and changes in social environment.

### 3.3. Characteristics of Study Variables and Inferential Statistics

Table 3 reveals significant differences in the mean scores for initiation and sustenance between individuals who had not engaged in drowsy driving in the previous month and those who had. Specifically, the average score for initiation was markedly higher for those who had not participated in drowsy driving (M = 2.80, SD = 1.37) compared to individuals who had engaged in drowsy driving (M = 1.76, SD = 1.27). Similarly, the sustenance mean score was significantly greater for individuals who had not engaged in drowsy driving (M = 3.01, SD = 1.19) in contrast to those who had (M = 1.52, SD = 1.14).

The hierarchical multiple regression analysis was employed to evaluate the effect of MTM constructs on the likelihood of initiating refraining from drowsy driving, by progressively incorporating the MTM constructs alongside covariates. This approach was aimed at determining the enhancement in predictive capability with the inclusion of MTM constructs (as shown in Table 4). The full model (Model 4), pertaining to individuals who had engaged in drowsy driving in the preceding month, demonstrated statistical significance with an adjusted R^2^ = 0.364 (*p* < 0.0001). This indicates that the full model accounts for 36.4% of the variance in the initiation of avoiding drowsy driving behaviors among students, which is a 31.4% increase in variance explained over the baseline model (Model 1). The VIF values for each variable in Model 4 were found to be under 10, suggesting an absence of multicollinearity among the variables. While changes in the physical environment did not significantly influence the likelihood of initiating avoidance of drowsy driving, substantial evidence was found supporting the positive impact of participatory dialogue and behavioral confidence on initiating such behavioral changes.

Using a hierarchical multiple regression approach like that for initiation, an analysis for the likelihood of sustaining behavior change was conducted. The results, presented in Table 5, reveal that the full model (Model 4) achieved an adjusted R^2^ value of 0.406 (*p* < 0.0001), marking a significant improvement over earlier models. This full model explains 35.2% more variance in the sustenance of behavior change compared to the baseline model (Model 1), indicating a substantial enhancement in predictive power. The VIF values for all included variables were found to be less than 10, which suggests there is no multicollinearity issue among the variables. Although changes in the social environment did not have a notable impact on the likelihood of maintaining the behavior change, strong evidence supports the significant roles of emotional transformation and the practice for change in promoting the sustenance of the behavioral change.

## 4. Discussion

In our study, we employed the multi-theory model (MTM) of health behavior change, an advanced fourth-generation behavior theory, to comprehensively explain drowsy driving behavior among university students. We found that 49.38% of the students reported drowsy driving in the past month, indicating that nearly half of the participants had engaged in this behavior. Previous studies have shown varying rates of drowsy driving among students, ranging from 16% [8] to 71.9% [10], with other studies reporting rates such as 20% [9] and 40%, to over 70% of students having experienced drowsy driving at least once in their lifetime [13]. Our study’s rate falls within this range, highlighting the need to develop interventions aimed at preventing drowsy driving behavior among college students.

The study demonstrated that 36.4% of the variance in the likelihood of initiating behavior to avoid drowsy driving can be accounted for by the final model. Within this model, only the constructs of participatory dialogue and behavioral confidence emerged as significant predictors. Together, these two constructs explained 31.2% of the variance in the likelihood of initiating the avoidance of drowsy driving. Participatory dialogue evaluates the pros and cons of drowsy driving, while behavioral confidence measures the certainty of abstaining from drowsy driving despite potential obstacles. These findings highlight the importance of participatory dialogue and behavioral confidence in interventions aimed at helping university students avoid drowsy driving.

Although this is the first application of the MTM to study drowsy driving behavior, previous MTM-based studies have demonstrated that participatory dialogue and behavioral confidence are effective and statistically significant predictors of behavior change initiation among college students [27,28,29,30]. For example, Sharma et al. (2016) found that these constructs, along with gender, explained 37% of the variance in the likelihood of initiating small portion size consumption behavior among college students [29]. A study by Sharma et al. (2020) revealed that 16.7% of the variance in the likelihood of initiating plain water consumption among college students could be explained by participatory dialogue and behavioral confidence [28]. Similarly, these constructs accounted for 27.2% of the variance in the likelihood of initiating handwashing behavior among college students [30]. Additionally, Olatunde et al. (2022) reported that 16.9% of the variance in the likelihood of initiating telehealth use among college students could be attributed to participatory dialogue and behavioral confidence [27].

The variance in the likelihood of sustaining the behavior change was 40.6% among those who participated in drowsy driving during the past month. Within this final model, two constructs of sustenance were found to be significant predictors. Those constructs were emotional transformation and practice for change. Together, these two constructs explained 35.2% of the variance in the likelihood of sustaining the avoidance of drowsy driving behavior. Emotional transformation involves converting feelings into goals of abstaining from drowsy driving and promoting continued abstinence of such behavior among college students. The practice of change construct focuses on creating a habit of abstaining from drowsy driving and integrating it into one’s way of life. These findings highlight the importance of incorporating emotional transformation and practice for change in interventions aimed at promoting long-term cessation of drowsy driving among college students. Several studies among college and university students have found that emotional transformation and practice for change constructs are effective and statistically significant predictors of maintaining behavior changes [31,32]. For instance, Sharma et al. (2017) found that these two constructs explained 58.3% of the variance in the likelihood of maintaining plain water consumption [32]. Similarly, Kapukotuwa et al. (2023) found that emotional transformation and practice for change accounted for 22.6% of the variance in the likelihood of maintaining quitting gambling behavior among university students [31].

Apart from the two sustenance constructs mentioned above, our findings indicate that the use of prescribed drugs is significantly and negatively associated with maintaining abstinence from drowsy driving behavior among college students. This aligns with Hershner and Chervin (2014), who primarily highlight the impact of stimulants and medications on sleep quality and their indirect influence on driving behaviors [14]. While this study suggests that the use of prescribed stimulants is associated with poor sleep quality and increased sleep latency, leading to higher risks of drowsy driving [14], they do not provide clear evidence that prescribed drug use directly affects the maintenance of abstinence from drowsy driving.

Our study also indicates that being a second- and third-year undergraduate student, being a graduate student, and being a professional degree student were found to be significantly and negatively associated with maintaining the abstinence of drowsy driving behavior compared to first-year undergraduate students. While previous studies on drowsy driving behavior among college and university students suggest that younger students are more likely to engage in drowsy driving [10,11], these studies do not provide clear evidence that being a second- and third-year undergraduate, graduate, or professional degree student is negatively associated with maintaining abstinence from drowsy driving. The data point more toward the prevalence of drowsy driving being higher among younger and less experienced drivers [10,11], which could imply better behavior among more advanced students. However, the specific focus on maintaining abstinence is less clear from the available studies. Our findings could be unique to our study group. To obtain a clearer understanding of our participants, future studies should investigate the role of academic standing and student status concerning drowsy driving behaviors more thoroughly, considering both abstinence and engagement aspects. This will help to determine if our results are consistent across different populations or if they are specific to our study group.

To our knowledge, this is the first study to examine the relationship between MTM and drowsy driving behavior. In this study, we did not find any significant relationship between changes in the physical environment and changes in the social environment constructs with drowsy driving. This lack of association can be attributed to the individualistic nature of drowsy driving behavior, which is primarily influenced by personal choices rather than external factors.

Support for this finding comes from several studies. For instance, Lindsay et al. (1999) highlighted the prevalence of dozing incidents among college students, emphasizing the role of individual sleep habits and personal choices rather than environmental factors [11]. Similarly, Hershner and Chervin (2014) discussed the impact of sleep deprivation and irregular sleep schedules on drowsy driving, focusing on individual sleep behaviors without significant mention of the influence of physical or social environments [14]. Robbins et al. (2021) further supported this by finding a strong link between poor sleep health and risky driving behaviors, attributing drowsy driving to personal sleep patterns and choices rather than external environmental changes [9]. Additionally, Lee et al. (2016) identified individual sleep patterns and behaviors as key factors influencing drowsy driving, reinforcing the idea that personal factors are the primary influencers of this behavior [10].

These findings collectively suggest that drowsy driving is predominantly driven by individual factors, with changes in physical and social environments playing a less significant role.

### 4.1. Implications for Practice

This study’s findings offer insightful information for creating focused interventions to help college and university students stop driving while drowsy. The results show that almost 50% of college students drove while drowsy in the previous month. Adopting population-specific behavioral modification interventions at the individual level is crucial to solving this problem. These treatments can be provided via a variety of channels, such as motor vehicle departments and student wellness centers on campus. They can also be provided through social media, mobile apps, tiredness monitoring systems, instructional campaigns, and sleep hygiene education. The efficacy of such treatments can be assessed using a randomized controlled trial (RCT) methodology, which enables comparisons between MTM-based interventions and other theory- or knowledge-based methods. Our study shows that participatory dialogue, behavioral confidence, practice for change, and emotional transformation constructs are important in the initiation and maintenance of drowsy driving among college students.

Highlighting the benefits of staying alert, such as increased safety, reduced accident risk, better health, improved reaction times, and enhanced focus, can help students recognize the positive impact of avoiding drowsy driving. Addressing perceived inconveniences like taking breaks, potential lost time, or maintaining a regular sleep schedule through discussions and time management strategies can further motivate safer driving behaviors.

To build behavioral confidence, it is essential to provide practical strategies and skills. Educating students on recognizing signs of drowsiness, such as frequent yawning or drifting from the lane, and training them on effective countermeasures like taking breaks, using caffeine wisely, or planning naps, can empower them to stay alert. Positive reinforcement through testimonials and safe practice opportunities can further enhance their confidence in avoiding drowsy driving.

Creating opportunities to apply and refine new skills and habits is important. Encouraging students to set realistic goals, like planning rest stops during long trips, maintaining a consistent sleep schedule, and using alertness-monitoring apps, can help integrate these practices into their routines. Keeping a driving journal to track alertness and recording the effectiveness of different countermeasures can help reflect on progress and identify areas for improvement.

Finally, achieving emotional transformation involves connecting behavior change to deeply felt personal values and emotions. Encouraging students to reflect on the emotional and psychological impact of drowsy driving, sharing stories of those affected by its consequences, and promoting positive emotional associations with staying alert can create powerful motivation for change. Celebrating small successes, like completing a long drive without feeling drowsy, can foster a sense of accomplishment and reinforce the commitment to safe driving habits.

### 4.2. Strengths and Limitations

This study employed the fourth-generation theory of health behavior change called the MTM, which is a novel approach that has been extensively studied and proven effective in the fields of health promotion and public health research. The MTM has demonstrated its versatility in facilitating behavior change across various health domains. A systematic review of studies from 2016 to 2023 highlights the MTM’s application across diverse behaviors, including smoking cessation, dietary improvements, physical activity, and more, demonstrating its broad effectiveness in predicting and promoting health behavior change [33].

The results of this study can facilitate the development of potential interventions utilizing the MTM to assist college and university students abstaining from drowsy driving behavior. The instrument utilized in this study exhibited acceptable readability, validity, and reliability, indicating its appropriateness for future cross-sectional and interventional studies. The SEM results also suggested that the instrument accurately measured the intended constructs. This indicates that researchers can confidently employ the instrument in future studies to evaluate drowsy driving behavior and its associated constructs among college and university students.

This study has several limitations. Firstly, the sample consisted only of college-aged individuals from a southwestern university, limiting the generalizability of the findings. Future research should include a more diverse range of universities and a larger sample size to enhance external validity. Secondly, reliance on self-reported data introduces biases such as recall bias and social desirability bias, though efforts were made to ensure participant confidentiality and anonymity. Thirdly, the study only examined behavior over the past 30 days, which may not reflect typical behavior. Fourthly, the lack of a test–retest reliability assessment means the instrument’s consistency over time was not evaluated, presenting an opportunity for future research to address this. Lastly, the cross-sectional design limits the ability to establish causality, and future longitudinal or experimental studies are needed to explore causal relationships in drowsy driving behavior among college students.

### 4.3. Recommendations

Drivers should avoid drinking alcohol before driving, obtain enough sleep (7–8 h for adults), and seek treatment for sleep disorders to minimize drowsy driving. The effects of alcohol, even in tiny doses, can worsen sleepiness-related driving impairment [34]. When driving, one should be aware of the signs of fatigue and react accordingly. Organizing regular workshops and seminars can educate students about the dangers of drowsy driving, using real-life stories and statistics to highlight the risks. Additionally, creating an online portal with educational materials such as videos, articles, and quizzes can further raise awareness. By implementing these recommendations, we can significantly reduce the incidence of drowsy driving among students, enhancing their safety and well-being. We are confident that with collective effort and commitment, we can make a substantial difference.

## 5. Conclusions

In conclusion, this study demonstrates the importance of using the multi-theory model (MTM) of health behavior change to address the issue of drowsy driving among college students. Addressing drowsy driving through evidence-based strategies could lead to better outcomes and contribute to improved road safety. This study yielded noteworthy results concerning the frequency of drowsy driving behavior among college students, indicating that about 50% of the participants engaged in drowsy driving in the past month. These tendencies give rise to concerns regarding the potential hazards of driving fatalities and road traffic accidents among this specific group. Significantly, this study is groundbreaking in its utilization of the novel theoretical framework, the MTM, to comprehend and elucidate drowsy driving behavior among college students. The results demonstrate the potential of the innovative MTM framework in revealing factors linked to many behaviors about the cessation or adoption of certain behaviors among college students. These findings indicate that future researchers can utilize the MTM framework to create focused interventions to help college and university students abstain from drowsy driving and any such potential behaviors contributing toward health and safety.

## Figures and Tables

**Figure 1 ijerph-21-01157-f001:**
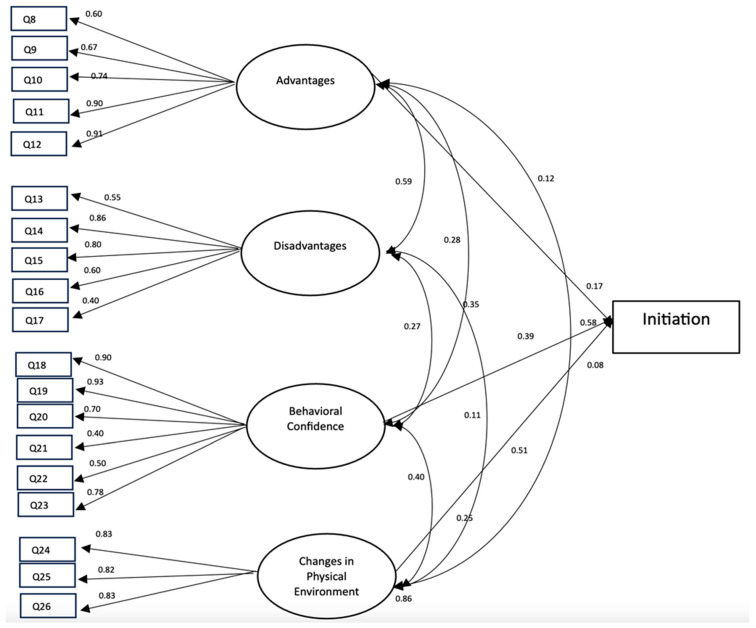
Structural equation modeling for initiation of safe driving behavior. Note: only statistically significant parameter estimates are displayed for conceptual clarity.

**Figure 2 ijerph-21-01157-f002:**
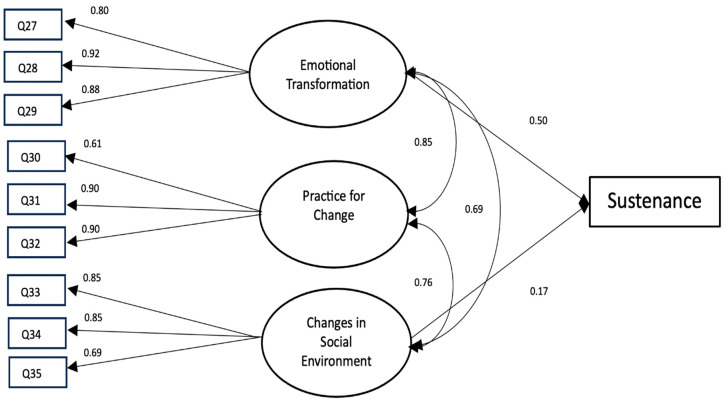
Structural equation modeling for sustenance of safe driving behavior. Note: only statistically significant parameter estimates are displayed for conceptual clarity.

**Table 1 ijerph-21-01157-t001:** Descriptive statistics of the demographic variables (n = 725).

Variable	Characteristics	Mean ± SD	n (%)
Age	-	26.01 ± 9.20	-
Gender	Female	-	480 (66.21)
	Male	-	216 (29.79)
	Other	-	29 (4.00)
Race/Ethnicity	White	-	270 (37.24)
	Hispanic, Latino/a, or Latinx	-	178 (24.55)
	Asian or Asian American	-	132 (18.21)
	Other races	-	145 (20.00)
Class	First year	-	126 (17.38)
	Second year	-	78 (10.76)
	Third year	-	116 (16.00)
	Fourth year		110 (15.17)
	Fifth year or more	-	50 (6.90)
	Graduate	-	204 (28.14)
	Professional	-	41 (5.66)
Drowsy driving behavior during the past month	Yes	-	358 (49.38)
	No	-	367 (50.62)
Alcohol consumption	Yes	-	330 (45.52)
	No	-	395 (54.48)
Driving under the influence status	Yes	-	173 (23.86)
	No	-	552 (76.14)
Usage of prescribed drugs	Yes	-	46 (6.34)
	No	-	679 (93.66)
Nightshift working status	Yes	-	174 (24.00)
	No	-	551 (76.00)
Sleeping disorder status	Yes	-	117 (16.14)
	No	-	608 (83.86)
Participation in late-night parties	Yes	-	200 (27.59)
	No	-	525 (72.41)
Employment	Yes	-	545 (75.17)
	No	-	180 (24.83)
Working hours per week *	-	28.64 ± 14.22	

* Only 545 (75.17%) students reported employment.

**Table 2 ijerph-21-01157-t002:** Internal consistency of the initiation and sustenance scales and subscales.

Scale	Cronbach’s Alpha (95% CI)
Perceived Advantage	0.88 (0.86, 0.91)
Perceived Disadvantage	0.78 (0.75, 0.81)
Behavioral Confidence	0.81 (0.79, 0.84)
Changes in the Physical Environment	0.87 (0.84, 0.89)
Overall Initiation Scale	0.86 (0.83, 0.89)
Emotional Transformation	0.90 (0.88, 0.92)
Practice for Change	0.83 (0.80, 0.86)
Changes in the Social Environment	0.83 (0.80, 0.86)
Overall Sustenance Scale	0.92 (0.90, 0.94)
Overall Scale	0.93 (0.91, 0.95)

**Table 3 ijerph-21-01157-t003:** Descriptive statistics of the multi-theory model of behavior change constructs (n = 725).

	Students Who Had Not Participated in Drowsy Driving in the Past Month (n = 367)			Students Who Participated in Drowsy Driving in the Past Month (n = 358)			
Constructs	Possible Score Range	Observed Score Range	Mean ± SD	Possible Score Range	Observed Score Range	Mean ± SD	*p*-Value
Initiation	0–4	0–4	2.80 ± 1.37	0–4	0–4	1.76 ± 1.27	<0.0001
Perceived Advantage (PA)	0–20	0–20	15.10 ± 5.34	0–20	0–20	13.30 ± 4.62	<0.0001
Perceived Disadvantage (PDA)	0–20	0–20	11.28 ± 5.34	0–20	0–20	9.45 ± 4.28	<0.0001
Participatory Dialogue (PA-PDA)	−20–+20	−14–+20	3.82 ± 4.88	−20–+20	−9–+16	3.84 ± 4.62	0.9471
Behavioral confidence	0–24	0–24	18.13 ± 5.35	0–24	0–24	11.07 ± 5.69	<0.0001
Changes in the physical environment	0–12	0–12	7.57 ± 3.87	0–12	0–12	6.26 ± 3.86	<0.0001
Sustenance	0–4	0–4	3.01 ± 1.19	0–4	0–4	1.52 ± 1.14	<0.0001
Emotional transformation	0–12	0–12	8.96 ± 3.07	0–12	0–12	6.17 ± 3.27	<0.0001
Practice for change	0–12	0–12	7.23 ± 3.29	0–12	0–12	4.88 ± 2.99	<0.0001
Changes in the social environment	0–12	0–12	7.74 ± 3.53	0–12	0–12	4.87 ± 3.36	<0.0001

**Table 4 ijerph-21-01157-t004:** Hierarchical multiple regression results for the initiation scale.

Variables		Model 1		Model 2		Model 3		Model 4	
		β	*p*-value	β	*p*-value	β	*p*-value	β	*p*-value
Intercept		2.141	<0.0001	1.852	<0.0001	0.214	0.4743	0.143	0.6368
Age		−0.008	0.4126	−0.006	0.5349	0.001	0.9189	−0.001	0.9460
Sex	(Female reference)								
	Male	−0.366	0.0165	−0.353	0.0180	−0.228	0.0682	−0.226	0.0712
Race or Ethnicity	(White reference)								
	Hispanic	0.357	0.0382	0.320	0.0576	0.089	0.5321	0.097	0.4935
	Asian	0.180	0.3718	0.082	0.6794	−0.086	0.6050	−0.071	0.6692
	Others	0.249	0.1826	0.238	0.1924	−0.082	0.5965	−0.092	0.5550
Class	(First year reference)								
	Second year	0.263	0.3140	0.238	0.3525	0.159	0.4585	0.159	0.4578
	Third year	−0.171	0.4835	−0.172	0.4708	−0.023	0.9071	0.004	0.9852
	Fourth year	0.179	0.4700	0.118	0.6269	0.205	0.3139	0.227	0.2655
	Fifth year or more	0.048	0.8691	0.047	0.8689	0.245	0.3101	0.281	0.2464
	Graduate	−0.137	0.5943	−0.148	0.5576	−0.011	0.9596	0.016	0.9413
	Professional	0.236	0.5142	0.234	0.5099	−0.013	0.9651	−0.030	0.9198
Alcohol consumption		−0.388	0.0144	−0.367	0.0183	−0.155	0.2360	−0.167	0.2013
DUI status		−0.092	0.5936	−0.092	0.5858	−0.150	0.2912	−0.145	0.3068
Prescribed drug use		−0.519	0.0328	−0.462	0.0527	−0.338	0.0900	−0.367	0.0671
Night shift working status		−0.190	0.2256	−0.210	0.1700	0.087	0.5071	0.080	0.5375
Sleeping disorder status		−0.141	0.4060	−0.039	0.8173	0.119	0.3968	0.132	0.3502
Nighttime partying status		0.117	0.4656	0.132	0.4011	0.090	0.4946	0.095	0.4720
Unemployment status		0.063	0.7177	0.090	0.5985	0.069	0.6294	0.075	0.6026
Participatory dialogue				0.057	<0.0001	0.042	0.0005	0.041	0.0008
Behavioral confidence						0.124	<0.0001	0.121	<0.0001
Changes in the physical environment								0.021	0.1703
R^2^		0.098		0.138		0.398		0.401	
ΔR^2^				0.040		0.260		0.003	
F		2.05	0.0074	2.84	<0.0001	11.13	<0.0001	10.72	<0.0001

Adjusted R^2^ of Model 4 (Full model) = 0.364.

**Table 5 ijerph-21-01157-t005:** Hierarchical multiple regression results for the sustenance scale.

Variables		Model 1		Model 2		Model 3		Model 4	
		β	*p*-value	β	*p*-value	β	*p*-value	β	*p*-value
Intercept		2.227	<0.0001	0.542	0.0371	0.454	0.0791	0.426	0.1013
Age		−0.006	0.5403	0.005	0.4806	0.006	0.4169	0.006	0.4174
Sex	(Female reference)								
	Male	−0.275	0.0450	−0.195	0.0766	−0.200	0.0663	−0.201	0.0653
Race or Ethnicity	(White reference)								
	Hispanic	0.138	0.3707	−0.019	0.8755	−0.032	0.7968	−0.031	0.8023
	Asian	0.013	0.9428	−0.026	0.8590	−0.046	0.7487	−0.055	0.7042
	Others	0.143	0.3942	0.038	0.7758	0.024	0.8597	0.016	0.9039
Class	(First year reference)								
	Second year	−0.106	0.6523	−0.192	0.3081	−0.182	0.3304	−0.168	0.3681
	Third year	−0.502	0.0225	−0.482	0.0063	−0.448	0.0104	−0.443	0.0114
	Fourth year	−0.337	0.1301	−0.388	0.0303	−0.360	0.0421	−0.356	0.0444
	Fifth year or more	−0.228	0.3880	−0.286	0.1769	−0.259	0.2169	−0.231	0.2744
	Graduate	−0.488	0.0357	−0.416	0.0255	−0.402	0.0289	−0.387	0.0360
	Professional	−0.203	0.5329	−0.541	0.0397	−0.553	0.0336	−0.570	0.0288
Alcohol consumption		−0.152	0.2871	−0.002	0.9869	−0.017	0.8792	−0.019	0.8671
DUI status		−0.248	0.1124	−0.211	0.0928	−0.212	0.0865	−0.217	0.0795
Prescribed drug use		−0.587	0.0074	−0.445	0.0114	−0.412	0.0181	−0.404	0.0206
Night shift working status		−0.253	0.0727	−0.149	0.1896	−0.165	0.1407	−0.164	0.1447
Sleeping disorder status		−0.205	0.1796	−0.030	0.8040	−0.021	0.8642	−0.019	0.8725
Nighttime partying status		0.110	0.4451	−0.022	0.8491	−0.006	0.9553	−0.011	0.9240
Unemployment status		0.022	0.8889	0.105	0.4077	0.102	0.4167	0.102	0.4131
Emotional transformation				0.210	<0.0001	0.169	<0.0001	0.166	<0.0001
Practice for change						0.065	0.0029	0.055	0.0202
Changes in the social environment								0.018	0.3121
R^2^		0.102		0.424		0.439		0.441	
ΔR^2^				0.322		0.015		0.002	
F		2.14	0.0048	13.12	<0.0001	13.20	<0.0001	12.63	<0.0001

Adjusted R^2^ of Model 4 (Full model) = 0.406.

## Data Availability

The data presented in this study are available upon request from the corresponding author. The data are not publicly available due to the presence of ethical reasons.

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
