# Peer review of "Theory-Based Determinants of Stopping Drowsy Driving Behavior in College Students: A Cross-Sectional Study"

_ijerph, 2024, doi:10.3390/ijerph21091157_

Round 1
Reviewer 1 Report
Comments and Suggestions for Authors
The article entitled "Theory-based determinants of stopping drowsy driving behavior in college students: A cross-sectional study" aimed to investigate the determinants of stopping drowsy driving behavior among college students using the multi-theory model (MTM) of health behavior change. The manuscript has been prepared carefully, and the following are a few suggestions that could be taken into account by the Authors to complete the paper:
- In the Introduction section, it would be useful to briefly expand on the idea of why, as the Authors stated, "The Theory of Planned Behavior and the Prototype Willingness Model have been suggested as useful frameworks for understanding and addressing drowsy driving" (lines 104-106).
- It seems desirable to highlight in the Introduction section the innovative nature of the work.
- It is recommended that more details be added regarding study participants in the Materials and Methods section, i.e., for example, age range, average age in the study group, number of women and men, even roughly to approximate the specifics of the study sample.
I hope that the above-mentioned suggestions will be helpful in improving the work, which, in my opinion, is both interesting and thoroughly prepared.
Best regards,
Reviewer
Author Response
The article entitled "Theory-based determinants of stopping drowsy driving behavior in college students: A cross-sectional study" aimed to investigate the determinants of stopping drowsy driving behavior among college students using the multi-theory model (MTM) of health behavior change. The manuscript has been prepared carefully, and the following are a few suggestions that could be taken into account by the Authors to complete the paper:
Thank you for sharing your time and your kind comments to strengthen our paper.
Point 1:
- In the Introduction section, it would be useful to briefly expand on the idea of why, as the Authors stated, "The Theory of Planned Behavior and the Prototype Willingness Model have been suggested as useful frameworks for understanding and addressing drowsy driving" (lines 104-106).
Response 1: Thank you for this astute suggestion. We modified and expanded it as follows (lines 104-113),
The Theory of Planned Behavior (TPB) and the Prototype Willingness Model (PWM) have been suggested as useful frameworks for understanding and addressing drowsy driving because they offer insights into the factors that influence individuals' intentions and willingness to engage in this risky behavior [10]. The TPB helps to explain how attitudes, perceived social pressures (subjective norms), and perceived control over the behavior contribute to one's intention to drive while drowsy. The PWM adds an additional layer by considering the role of social prototypes and situational factors that can influence individuals to engage in drowsy driving even when they have no prior intention to do so. Together, these models provide a comprehensive approach to predicting and modifying drowsy driving behaviors [10].
We hope the revisions will meet with your kind approval.
Point 2:
- It seems desirable to highlight in the Introduction section the innovative nature of the work.
Response 2: Thank you for this astute suggestion. We added the following section to the introduction (lines 132-142),
The innovative aspect of this study lies in the application of the MTM to drowsy driving behavior among college students. To the best of our knowledge, this is the first study to utilize the MTM in exploring this behavior within this population. By employing MTM, this study not only addresses a critical public health concern but also demonstrates the versatility and applicability of this novel theoretical model in understanding and promoting behavior change in real-world settings. The MTM's comprehensive approach offers new insights into both the initiation and sustenance of behavior change, making it a powerful tool for developing effective interventions. Thus, the purpose of this study is to elucidate the extent to which the MTM constructs of initiation and sustenance explain the cessation of drowsy driving behavior among college students at a large Southwestern University.
We hope the revisions will meet with your kind approval.
Point 3:
- It is recommended that more details be added regarding study participants in the Materials and Methods section, i.e., for example, age range, average age in the study group, number of women and men, even roughly to approximate the specifics of the study sample.
Response 3: Thank you for your kind suggestion. The demographic information are presented in the results section and more details can be found in the Table 1. Therefore, we do not think it is necessary to add these information to the Materials and Methods section. Hope you will agree with our rationale.
I hope that the above-mentioned suggestions will be helpful in improving the work, which, in my opinion, is both interesting and thoroughly prepared.
Best regards,
Reviewer
Thank you once again for strengthening our paper by providing invaluable comments and sharing with us your acumen and time.
Reviewer 2 Report
Comments and Suggestions for Authors
The work reported in this paper is very interesting and very timely. The research is potentially important for two reasons: (1) because it applies the multi-theory model of health behavior change (MTM) -- and illustrates how this can be done, and (2) because the application domain upon which it focuses is an important one (drowsy driving, one of several driver states that is prioritized for research and action by the National Highway Traffic Safety Administration at the current time. Behavior-change is increasingly important today – not just for transportation, but in many domains – from pandemic-responses to transportation – and to the extent that emerging theoretical frameworks (like the MTM) can provide improved understanding and better guidance for developing interventions, they will be important. And relative to the topic of drowsy driving – among college-aged drivers – it has been the case that many drive in spite of being aware that they are drowsy – and new insights into why this is the case (and what can be done about it) are very important as the US and other countries around the globe seek to reduce fatalities and crashes.
Furthermore, this research is very well-conceived and very well- communicated. The conceptual and theoretical background is covered clearly and cogently – and the structure and composition of the survey instrument that the authors have developed on the basis of the MTM is very clearly described, along with its psychometric evaluation (which has been thoughtfully and carefully attended to). The survey study that was conducted is well-described, and the method carefully implemented. Many aspects of the data analyses are clear – and the discussion is similarly well-done.
For these reasons, I recommend accepting the paper for publication – if three minor revisions are made first. These three minor revisions are described below.
1. (1) Add one or two sentences to the end of the introduction to explain the logic or rationale for predicting the initiation and sustenance of behavior change in the absence of any program to change the behavior (or to sustain the changed behavior). If the notion of the study was to try to obtain a “baseline” picture of the MTM scores for those students who did *not* drive drowsy on the determinants associated with "initiating" and "sustaining" compared to those students who did drive drowsy . . . then it would be helpful to state that simply, straightforwardly, and explicitly at the beginning. If the central notion was to see if the scores of these two groups of students differed significantly on the two subsets of MTM scores even before any special intervention programs were administered – and, if they did, then it would provide support for the MTM framework – and might furthermore help guide development of the content and focus of future intervention programs to prevent drowsy driving. I think that adding just a small straightforward couple of sentences would help the reader a great deal.
One or two sentences of this type could be added near the end of the Introduction – at around Line 126, where there is a sentence explaining that “the purpose of the study was to elicit the extent to which the MTM constructs of initiation and sustenance explained stopping the drowsy driving behavior of college students.” I think that right there an addition could be added to say something like . . . “even in the absence of any special program or intervention encouraging them to stop.” And then one or two additional sentences could be added to say that . . . “By comparing the MTM scores for students who drove while drowsy to those who refrained from driving while drowsy, the objective of this study was to obtain a baseline understanding of how well the MTM differentiated these behaviors – even before any behavior-change interventions were introduced. If the MTM is successful in differentiating them in this baseline circumstance, it would suggest that it can offer real promise for guiding intervention development in the future. (Or something of that sort.)
Note: I suggest this because during my first time reading the paper, I became confused in the analysis section about whether there was a behavior-change for which a “sustenance” outcome was measured over some period of time -- or whether some intervention had occurred -- had led to a behavior-change -- and then sustenance was measured over a period of time. It took me awhile to figure out that there was just a “behavior” ("driving drowsy" – or "not driving drowsy") – and no behavior-change program -- and the scores on the survey instrument were simply computed and compared for those with the behavior vs. those without it. The addition suggested above would have made it easier for me to understand the paper -- and would have prevented me from becoming confused during the analysis section. So I hope it will help other readers understand the whole paper pretty seamlessly.
(2) In either the methods section or the analysis section, I’d suggest that you add one brief sentence or two to explain that, in addition to Hierarchical Multiple Regression, you used Structural Equation Modeling to visualize the latent variables and illustrate the statistically-significant parameter estimates (or whatever explanation you feel offers the best and most accurate explanation from your point of view). That way, your readers will feel prepared and supported when they encounter Figure 1 and 2.
(3) In the discussion section somewhere – maybe under “strengths” – I would encourage you to mention that the MTM has applicability to other types of behavior-change – beyond drowsiness & transportation -- and give an example or two. [I, for one, am pleased to see some advances beyond the Theory of Planned Behavior, and appreciate the incorporation of emotion and feelings, as well as supportive changes in the physical and social environment, and some of the other newer elements (like practice).]
Very nicely done research; it could have significant impact on the behavior-change work going on now in the field.
Author Response
The work reported in this paper is very interesting and very timely. The research is potentially important for two reasons: (1) because it applies the multi-theory model of health behavior change (MTM) -- and illustrates how this can be done, and (2) because the application domain upon which it focuses is an important one (drowsy driving, one of several driver states that is prioritized for research and action by the National Highway Traffic Safety Administration at the current time. Behavior-change is increasingly important today – not just for transportation, but in many domains – from pandemic-responses to transportation – and to the extent that emerging theoretical frameworks (like the MTM) can provide improved understanding and better guidance for developing interventions, they will be important. And relative to the topic of drowsy driving – among college-aged drivers – it has been the case that many drive in spite of being aware that they are drowsy – and new insights into why this is the case (and what can be done about it) are very important as the US and other countries around the globe seek to reduce fatalities and crashes.
Furthermore, this research is very well-conceived and very well- communicated. The conceptual and theoretical background is covered clearly and cogently – and the structure and composition of the survey instrument that the authors have developed on the basis of the MTM is very clearly described, along with its psychometric evaluation (which has been thoughtfully and carefully attended to). The survey study that was conducted is well-described, and the method carefully implemented. Many aspects of the data analyses are clear – and the discussion is similarly well-done.
Thank you for sharing your time and your kind comments to strengthen our paper.
For these reasons, I recommend accepting the paper for publication – if three minor revisions are made first. These three minor revisions are described below.
Point 1:
Add one or two sentences to the end of the introduction to explain the logic or rationale for predicting the initiation and sustenance of behavior change in the absence of any program to change the behavior (or to sustain the changed behavior). If the notion of the study was to try to obtain a “baseline” picture of the MTM scores for those students who did *not* drive drowsy on the determinants associated with "initiating" and "sustaining" compared to those students who did drive drowsy . . . then it would be helpful to state that simply, straightforwardly, and explicitly at the beginning. If the central notion was to see if the scores of these two groups of students differed significantly on the two subsets of MTM scores even before any special intervention programs were administered – and, if they did, then it would provide support for the MTM framework – and might furthermore help guide development of the content and focus of future intervention programs to prevent drowsy driving. I think that adding just a small straightforward couple of sentences would help the reader a great deal.
One or two sentences of this type could be added near the end of the Introduction – at around Line 126, where there is a sentence explaining that “the purpose of the study was to elicit the extent to which the MTM constructs of initiation and sustenance explained stopping the drowsy driving behavior of college students.” I think that right there an addition could be added to say something like . . . “even in the absence of any special program or intervention encouraging them to stop.” And then one or two additional sentences could be added to say that . . . “By comparing the MTM scores for students who drove while drowsy to those who refrained from driving while drowsy, the objective of this study was to obtain a baseline understanding of how well the MTM differentiated these behaviors – even before any behavior-change interventions were introduced. If the MTM is successful in differentiating them in this baseline circumstance, it would suggest that it can offer real promise for guiding intervention development in the future. (Or something of that sort.)
Note: I suggest this because during my first time reading the paper, I became confused in the analysis section about whether there was a behavior-change for which a “sustenance” outcome was measured over some period of time -- or whether some intervention had occurred -- had led to a behavior-change -- and then sustenance was measured over a period of time. It took me awhile to figure out that there was just a “behavior” ("driving drowsy" – or "not driving drowsy") – and no behavior-change program -- and the scores on the survey instrument were simply computed and compared for those with the behavior vs. those without it. The addition suggested above would have made it easier for me to understand the paper -- and would have prevented me from becoming confused during the analysis section. So I hope it will help other readers understand the whole paper pretty seamlessly.
Response 1: Thank you for your astute observation. As suggested, we have added the following (lines 142-148),
This study compared MTM scores between students who drove while drowsy and those who did not, without any special program or intervention to stop drowsy driving. The objective of this was to obtain a baseline understanding of how well the MTM differentiated these behaviors – even before any behavior-change interventions are introduced. It was presumed that if the MTM is successful in differentiating them in this baseline circumstance, it would suggest that it can offer real promise for guiding intervention development in the future.
We hope this will meet your expectations.
Point 2:
In either the methods section or the analysis section, I’d suggest that you add one brief sentence or two to explain that, in addition to Hierarchical Multiple Regression, you used Structural Equation Modeling to visualize the latent variables and illustrate the statistically-significant parameter estimates (or whatever explanation you feel offers the best and most accurate explanation from your point of view). That way, your readers will feel prepared and supported when they encounter Figure 1 and 2.
Response 2: Thank you for your suggestion. We edited the analysis section as follows. (lines 221-234)
To evaluate internal consistency of our scales, Cronbach’s alpha coefficients were computed for both the entire scale and its constituent subscales, which are delineated by specific constructs, to evaluate internal consistency. An alpha value of 0.70 was established as the minimum acceptable limit, signifying that the scale and its sub-scales possess an adequate level of reliability in measuring the constructs they are intended to assess.
To further explore reliability, unidimensionality and validity, each scale was modeled as a latent variable using common factor analysis in Mplus. To test the con-struct validity, the scales turned factors were correlated, representing a measurement model. This was implemented using Mplus. The combination of the following fit indices was utilized to measure the degree of overall fit of the measurement model to data: comparative fit index (CFI) and Tucker - Lewis Index (TLI) with the values above 0.90 [23], and the root mean square error of approximation (RMSEA) and Standardized Root Mean Square Residual (SMRM) less than 0.08 [24]. We used the 0.10, 0.30, and .050 guidelines to explain effect size for identifying small, medium, and large effects, respectively [25].
Furthermore, we added “and structural equation modeling were” to the following sentence (line 244)
Hierarchical multiple regression and structural equation modeling were employed to predict the initiation and sustenance of quitting drowsy driving behavior.
We hope this will meet your expectations.
Point 3:
In the discussion section somewhere – maybe under “strengths” – I would encourage you to mention that the MTM has applicability to other types of behavior-change – beyond drowsiness & transportation -- and give an example or two. [I, for one, am pleased to see some advances beyond the Theory of Planned Behavior, and appreciate the incorporation of emotion and feelings, as well as supportive changes in the physical and social environment, and some of the other newer elements (like practice).]
Response 3: Thank you for this astute suggestion. We added following to the strengths and limitations section (lines 486-491)
The MTM has demonstrated its versatility in facilitating behavior change across various health domains. A systematic review of studies from 2016 to 2023 highlights MTM’s application across diverse behaviors, including smoking cessation, dietary improvements, physical activity, and more, demonstrating its broad effectiveness in predicting and promoting health behavior change [33].
We hope this will meet your expectations.
Very nicely done research; it could have significant impact on the behavior-change work going on now in the field.
Thank you once again for strengthening our paper by providing invaluable comments and sharing with us your acumen and time.